# Dendritic Cells Pulsed with Tumor Lysates Induced by Tetracyanotetra(aryl)porphyrazines-Based Photodynamic Therapy Effectively Trigger Anti-Tumor Immunity in an Orthotopic Mouse Glioma Model

**DOI:** 10.3390/pharmaceutics15102430

**Published:** 2023-10-06

**Authors:** Tikhon S. Redkin, Ekaterina E. Sleptsova, Victoria D. Turubanova, Mariia O. Saviuk, Svetlana A. Lermontova, Larisa G. Klapshina, Nina N. Peskova, Irina V. Balalaeva, Olga Krysko, Tatiana A. Mishchenko, Maria V. Vedunova, Dmitri V. Krysko

**Affiliations:** 1Institute of Neurosciences, National Research Lobachevsky State University of Nizhny Novgorod, 23 Gagarin Ave., 603022 Nizhny Novgorod, Russia; big.t.nsdav@outlook.com (T.S.R.); ees222@list.ru (E.E.S.); mariia.saviuk@ugent.be (M.O.S.); 2Cell Death Investigation and Therapy Laboratory, Anatomy and Embryology Unit, Department of Human Structure and Repair, Faculty of Medicine and Health Sciences, Ghent University, 9000 Ghent, Belgium; olga.krysko@ugent.be; 3Cancer Research Institute Ghent, 9000 Ghent, Belgium; 4Sector of Chromophors for Medicine, G.A. Razuvaev Institute of Organometallic Chemistry of the Russian Academy of Sciences, 49 Tropinin St., 603137 Nizhny Novgorod, Russia; lermontovasa@rambler.ru (S.A.L.); klarisa@iomc.ras.ru (L.G.K.); 5Institute of Biology and Biomedicine, National Research Lobachevsky State University of Nizhny Novgorod, 23 Gagarin Ave., 603022 Nizhny Novgorod, Russia; nin-22@yandex.ru (N.N.P.); irin-b@mail.ru (I.V.B.); saharnova87@mail.ru (T.A.M.); mvedunova@yandex.ru (M.V.V.); 6Department of Pathophysiology, Sechenov First Moscow State Medical University (Sechenov University), 125009 Moscow, Russia

**Keywords:** ICD, photosensitizers, GL261, DAMPs, ATP, HMGB1, prophylactic tumor vaccination

## Abstract

Research in the past decade on immunogenic cell death (ICD) has shown that the immunogenicity of dying tumor cells is crucial for effective anticancer therapy. ICD induction leads to the emission of specific damage-associated molecular patterns (DAMPs), which act as danger signals and as adjuvants to activate specific anti-tumor immune responses, leading to the elimination of tumor cells and the formation of long-term immunological memory. ICD can be triggered by many anticancer treatment modalities, including photodynamic therapy (PDT). However, due to the variety of photosensitizers used and the lack of a universally adopted PDT protocol, there is a need to develop novel PDT with a proven ICD capability. In the present study, we characterized the abilities of two photoactive dyes to induce ICD in experimental glioma in vitro and in vivo. One dye was from the tetracyanotetra(aryl)porphyrazine group with 9-phenanthrenyl (**pz I**), and the other was from the 4-(4-fluorobenzyoxy)phenyl (**pz III**) group in the aryl frame of the macrocycle. We showed that after the photosensitizers penetrated into murine glioma GL261 cells, they localized predominantly in the Golgi apparatus and partially in the endoplasmic reticulum, providing efficient phototoxic activity against glioma GL261 cells upon light irradiation at a dose of 20 J/cm^2^ (λex 630 nm; 20 mW/cm^2^). We demonstrated that **pz I**-PDT and **pz III**-PDT can act as efficient ICD inducers when applied to glioma GL261 cells, facilitating the release of two crucial DAMPs (ATP and HMGB1). Moreover, glioma GL261 cells stimulated with **pz I**-PDT or **pz III**-PDT provided strong protection against tumor growth in a prophylactic subcutaneous glioma vaccination model. Finally, we showed that dendritic cell (DC) vaccines pulsed with the lysates of glioma GL261 cells pre-treated with **pz-I**-PDT or **pz-III**-PDT could act as effective inducers of adaptive anti-tumor immunity in an intracranial orthotopic glioma mouse model.

## 1. Introduction

Over the past decade, the concept of immunogenic cell death (ICD) has drawn much attention in the context of anti-tumor immunity and cancer therapy [1,2,3,4]. It has been shown that a key to the success of such a strategy is the triggering of ICD, which induces a specific anti-tumor immune response, leading to tumor regression, the complete elimination of tumor cells, and the formation of long-term immunological memory [5,6]. Cancer cells undergoing ICD contribute to the emission of immuno-stimulatory molecules, such as damage-associated molecular patterns (DAMPs), the most common of which are ATP, high-mobility group box 1 (HMGB1), and calreticulin (CRT) [3,6,7,8]. These DAMPs act as adjuvants for professional antigen-presenting cells (e.g., dendritic cells), facilitating their phagocytic activation and phenotypic maturation in the presence of dead/dying cancer cells that undergo ICD. On the other hand, dying cancer cells have to be antigenic to enable the induction of antigen-specific anti-tumor immunity [9,10] associated with the effective cross-presentation of immunogenic peptides on major histocompatibility complex class I (MHC I) to CD8^+^ T cells, which play a key role in the anti-tumor response [7,9] and the formation of long-term immunological memory [7,9]. Thus, both the adjuvanticity and antigenicity of dying cancer cells are required for ICD and for the successful induction of anti-tumor immunity.

Several anticancer strategies are currently known to efficiently induce ICD in different types of tumors [11,12,13,14]; one example is photodynamic therapy (PDT). PDT is a minimally invasive, clinically approved therapeutic modality based on the systemic administration of a photodynamic agent (i.e., photosensitizer) and irradiation with visible light of an appropriate wavelength after the photosensitizer accumulates in the tumor cells [15,16,17,18]. This stepwise procedure causes the activation of a cascade of reactions, leading to the development of cytotoxic effects of singlet oxygen and other toxic reactive oxygen species, which interact with intracellular components and cause oxidative damage, resulting in tumor tissue destruction and vascular damage. However, due to the variety of photosensitizers used and the lack of a universally adopted PDT protocol, cancer cell death can follow different regulated cell death modalities, with immunogenic or non-immunogenic properties [18,19,20]. Therefore, it is crucial to assess the effectiveness of novel photosensitizers and their photodynamic potential in PDT treatment by inducing ICD in vitro and in vivo.

This study analyzed the ability of PDT based on tetracyanotetra(aryl)porphyrazines dyes with 9-phenanthrenyl (**pz I**) and 4-(4-fluorobenzyoxy)phenyl (**pz III**) groups in the aryl frame of the macrocycle to induce ICD in murine glioma GL261 cells. In particular, we assessed the release of several DAMPs (ATP and HMGB1) from glioma GL621 cells after **pz I**-PDT or **pz III**-PDT in vitro and analyzed whether dying glioma GL261 cells induced by **pz I**- or **pz III**-based PDT can protect mice from tumor growth in a prophylactic subcutaneous glioma vaccination model. We also examined the effectiveness of dendritic cell (DC) vaccines pulsed with the lysates of glioma GL261 cells after triggering ICD with **pz I**-PDT or **pz III**-PDT for the induction of anti-tumor immunity in an intracranial orthotopic glioma model in immunocompetent C57Bl/6 mice.

## 2. Materials and Methods

### 2.1. Glioma Cell Line

Murine glioma GL261 cells were cultured in DMEM medium (PanEco, Moscow, Russia) supplemented with 4.5 g/L glucose (PanEco, Moscow, Russia), 2 mM L-glutamine (PanEco, Moscow, Russia), 100 μM sodium pyruvate (Thermo Fisher, Waltham, MA, USA), 100 units/mL penicillin (PanEco, Moscow, Russia), 100 μg/L streptomycin (PanEco, Moscow, Russia), and 10% fetal bovine serum (Thermo Fisher, Waltham, MA, USA). At the end of the exponential growth phase, the cells were detached with trypsin-versine solution (1:3), centrifuged at 1000 rpm for 3 min, and reseeded at a density of 5 × 10^5^ cells/mL and a multiplicity of seeding of 1:10. Incubation was maintained at 37 °C under 5% CO_2_ in humidified air in a Binder C150 incubator (BINDER GmbH, Tuttlingen, Germany). All the experiments were performed after the third passage [21].

### 2.2. Photodynamic Induction of Glioma Cell Death

Tetracyanotetra(aryl)porphyrazine dyes with 9-phenanthrenyl (**pz I**) and 4-(4-fluorobenzyoxy)phenyl (**pz III**) groups in the aryl frame of the macrocycle were used for photoinduction of cell death in glioma GL261 cells. The synthesis of **pz I** and **pz III** were performed according to a previously described approach [22,23,24,25]. Glioma GL261 cells were seeded at approximately 5 × 10^5^ cells per Petri dish (Ø60 mm, Techno Plastic Products, Trasadingen, Switzerland) and grown overnight (24 h). Next, the cells were incubated with 2.8 µM **pz I** or 1.7 µM **pz III** solution in serum-free medium for 4 h in a CO_2_ incubator with extra protection from light. The **pz**-containing medium was replaced by a complete growth medium, and the glioma cells were irradiated at 20 J/cm^2^ with an LED light source (λex 630 nm; 20 mW/cm^2^).

### 2.3. Subcellular Distribution of pz I and pz III in Glioma Cells

Glioma GL261 cells were seeded in 96-well glass-bottom plates (Corning Inc., Corning, NY, USA) at 1 × 10^4^ cells per well and grown overnight (24 h). Next, the glioma cells were incubated in a serum-free medium supplemented with 10 μM **pz I** or **pz III** for 4 h, followed by washing with phosphate-buffered saline (PBS, Gibco, Thermo Fisher Scientific, Waltham, MA, USA). To analyze the intracellular distribution of **pz I** and **pz III**, the following dyes were added 30 min before the end of the incubation period: LysoTracker Green DND-26 for lysosomes (0.5 μM), ERTracker for endoplasmic reticulum (ER, 0.5 μM), MitoTracker Green FM for mitochondria (0.5 μM), BODIPY FL C5-ceramide complexed with bovine serum albumin for Golgi apparatus (5 μM), and DAPI for nucleus (2.8 μM) (all fluorescent dyes were purchased from Thermo Fisher Scientific, Waltham, MA, USA). The stained glioma GL261 cultures were observed in an LSM 710 Axio Obzerver Z1 DUO NLO laser scanning microscope (Carl Zeiss, Oberkochen, Germany) with an LD C-Apochromat water immersion objective lens, 40×/1.1. Fluorescence of the stained organelles was excited using an argon laser at 488 nm and recorded at 500–560 nm [26].

### 2.4. Cell Death Assay via MTT and Flow Cytometry

The viability analyses of glioma GL261 cells were performed at the following time points after **pz I**-PDT and **pz III**-PDT: 1.5 h; 3 h; 14 h; 18 h; 24 h; or immediately after PDT.

For the MTT assay at the respective time points, the photoinduced glioma GL261 cells were incubated in a serum-free culture medium supplemented with 10% MTT reagent (PanEco, Moscow, Russia) for 2 h. The formazan crystals formed were dissolved in dimethyl sulfoxide (DMSO), and then optical density measurements of the solution were performed at a 570 nm wavelength using a Synergy MX Microplate Reader (BioTek Instruments Inc., USA) [27].

For the flow cytometry analysis, the photoinduced glioma GL261 cells were detached from Petri dishes using a trypsin-versine solution (1:3), centrifuged at 1000 rpm for 3 min, washed in Annexin V binding buffer, and stained with a fluorescent nuclear marker (DAPI; Thermo Fisher Scientific, Waltham, MA, USA) and Annexin V FITC (Invitrogen, Waltham, MA, USA). The samples obtained were run on a BD FACSCanto (BD, Becton, NJ, USA) flow cytometer and then analyzed with FlowJo software (v.10.0.8) [28].

### 2.5. Analysis of DAMP Release: HMGB1 and ATP

Quantitative analyses of ATP release from the photoinduced glioma GL261 cells were performed at the following time points: 1.5 h; 3 h; 14 h; 18 h; 24 h; or immediately after PDT (0 h). The cell-free supernatants were collected from Petri dishes and centrifuged at 1000 rpm for 3 min. ATP release was measured using a CellTiter-Glo Luminescent Cell Viability Assay Kit (G7571, Promega, Madison, WI, USA) in accordance with the manufacturer’s protocol and a Synergy MX Microplate Reader (BioTek Instruments Inc., Winooski, VT, USA) [27].

Quantitative analyses of HMGB1 release from the photoinduced glioma GL261 cells were performed at the following time points: 1.5 h; 3 h; 14 h; 24 h; or immediately after PDT (0 h). Cell-free supernatants were collected from Petri dishes, centrifuged at 15,000 rpm at 4 °C for 3 min, and then stored at −80 °C in a Binder UF V 700 ultra-low-temperature freezer (BINDER GmbH, Tuttlingen, Germany). The HMGB1 in the supernatants was measured using an ELISA kit (IBL-Hamburg, Hamburg, Germany) in accordance with the manufacturer’s instructions. The measurements were performed in a Synergy MX Microplate Reader (BioTek Instruments Inc., Winooski, VT, USA) [27].

### 2.6. Mouse Experiments

In vivo experiments were performed on female C57BL/6 mice (7–8 weeks old). The animals were housed in specific pathogen-free conditions in the vivarium of Lobachevsky State University of Nizhny Novgorod.

#### 2.6.1. Prophylactic Subcutaneous Glioma Vaccination Model

Glioma GL261 cells were exposed to **pz I**-PDT or **pz III**-PDT as described above and then cultured overnight (24 h) in a complete growth medium in a CO_2_ incubator. After that, supernatant with dead/dying glioma GL261 cells was collected from the Petri dishes, centrifuged at 1000 rpm for 3 min, and resuspended in PBS. The mice were inoculated subcutaneously twice in the left flank, one week apart, with PDT-treated glioma GL261 cells (5 × 10^5^ cells). The control mice were injected with the same quantity of glioma GL261 cells subjected to four freeze–thaw cycles in regimen of freezing at −80 °C and thawing at +55 °C, which leads to non-immunogenic necrotic cell death (F/T, negative control, non-ICD) [26,27,28], or they were injected with PBS only. On day 8, after the last immunization, the mice were challenged subcutaneously in the opposite flank with viable glioma GL261 cells (1 × 10^5^ cells). Tumor growth at the challenge site was measured using a caliper every 2 days for 28–32 days. The mice were euthanized if the tumor volume exceeded 2 cm^3^ or if necrotic changes were observed [27].

#### 2.6.2. Dendritic Cell Vaccines

Immature bone-marrow-derived dendritic cells (DCs) were isolated from the femurs and tibias of female C57BL/6 mice (7–10 weeks old) and cultured in RPMI 1640 medium (Gibco, Waltham, MA, USA) supplemented with 5% heat-inactivated fetal bovine serum (Gibco, Waltham, MA, USA), 20 ng/mL mouse granulocyte-macrophage colony-stimulating factor (GM-CSF, VIB Protein Core, IRC-UGent, Ghent, Belgium), 1% L-glutamine (Gibco, Waltham, MA, USA), 1 mM sodium pyruvate (Thermo Fisher, Waltham, MA, USA), 50 µM β-mercaptoethanol, 100 U/mL penicillin, and 100 µg/L streptomycin. The culture medium was refreshed on days 3 and 6.

To prepare the DC vaccine, glioma GL261 cells were exposed to **pz I**-PDT or **pz III**-PDT as described above. After 24 h, the photoinduced dead/dying glioma GL261 cells were collected and centrifuged at 3000 rpm for 3 min. The cells were then subjected to six cycles of freezing (−80 °C) and thawing (+55 °C) to obtain protein for further DC activation in vitro. Total protein in the cell lysate was measured with a commercial BCA Protein Assay Kit (Sigma-Aldrich, Darmstadt, Germany) using a Synergy MX Microplate Reader (BioTek Instruments Inc., Winooski, VT, USA), and 2 mg of protein was added for 90 min to a suspension of 10 × 10^6^ DCs on day 7 of their cultivation. The DCs were then additionally stimulated via incubation for 24 h with *E. coli* lipopolysaccharide (LPS) (0.5 μg/mL, Sigma Aldrich, Darmstadt, Germany). DC maturation was analyzed via flow cytometry with a BD FACSCanto (BD, Becton, NJ, USA). The data were analyzed with FlowJo software (v.10.0.8) [21].

#### 2.6.3. Prophylactic DC Vaccination in an Orthoptic Intracranial Glioma Mouse Model

DC vaccines based on photoinduced glioma GL261 cell lysates were prepared as described above (Section 2.6.2). The mice were immunized intraperitoneally twice in the left flank, one week apart, with a suspension of 10 × 10^6^ prepared DCs. On day 8, after the last vaccination, viable glioma GL261 cells were injected intracranially with 2 × 10^4^ cells per 20 g of mouse weight. Before injection, the mice were anesthetized with a mixture of medical oxygen and isoflurane (induction: 5%; maintenance: 2%) and immobilized in a stereotaxic frame. The viable glioma GL261 cells suspended in 2 µL of PBS were injected via stereotaxic coordinates (2 mm lateral and 2 mm posterior to the bregma, and 3 mm below the dura mater) according to our previous studies [18,21]. The operating window was then sutured with surgical thread (0.2 mm) and treated with an antiseptic solution. After recovery from anesthesia, the mice were returned to their cages with ad libitum access to food and water.

#### 2.6.4. Neurological Deficiency Assessment

The neurological status of mice after inoculation of viable glioma GL261 cells was determined by using a scale to assess the severity of neurological deficit in small laboratory animals, with modifications [8,18,21,29]. The scale consisted of 14 tests that comprehensively assessed the motor activity of mice, the trajectory and coordination of their movements, the severity of reflexes damage, the state of muscle tone, and the presence/absence of ptosis and exophthalmos. Each test was scored 2 points for no reaction, 0 for good/normal reaction, and 1 for some disturbances [21]. Then, the values were summarized, and damage to the central nervous system was interpreted as follows: 15–19 points, severe damage; 8–14 points, moderate damage; and 3–7 points, light damage.

#### 2.6.5. Magnetic Resonance Imaging

Magnetic resonance imaging (MRI) was performed using a high-field magnetic resonance tomograph (Agilent Technologies DD2–400 9.4 T 400 MHz; Cheadle, UK) with a volume coil M2M (H1). The mice were anesthetized intraperitoneally with 70 mg/kg Zoletil 100 (Virbac Sante Animale, Carros, France) and intramuscularly with 0.02 mg/kg Xylanite (NITA-Pharm, Saratov, Russia) and then fixed in an upright position inside the magnet tunnel set at 27 °C.

The VnmrJ software was used for brain scanning and data processing. T1 tomograms of the layered frontal brain sections weighted by proton density were obtained using the multi-gradient echo multi slice (MGEMS) pulse sequence with the following parameters: TR = 1000 ms; TE = 1.49 ms; 6 echoes; FOV—20 × 20 mm; matrix—256 × 256; 15 slices; 1 mm slice thickness; 17 min; and 4 s scanning time [21].

### 2.7. Statistical Analysis

The data were analyzed with Mann–Whitney test in GraphPad Prism 9 software (San Diego, CA, USA). Kaplan–Meier survival curves, representing the time course of tumor growth, were analyzed using the log-rank Mantel–Cox test. Differences between tumor volumes were analyzed using the multiple *t*-test. The data were considered to be statistically significant if *p* < 0.05. At least three independent biological replicates were performed in all the experiments.

## 3. Results

### 3.1. pz-I-PDT and pz III-PDT Effectively Induce ICD in Glioma GL261 Cells In Vitro

#### 3.1.1. Analysis of Subcellular Localization of pz I and pz III in Glioma Cells

The localization of the photosensitizer in cellular compartments plays a significant role in determining the primary targets of PDT and the subsequent course of events, including the molecular mechanisms of cell death pathways and the ability to induce ICD [16,30,31,32,33,34]. We have previously shown that **pz I** and **pz III** actively accumulate in glioma GL261 cells within 2 h of incubation, providing the conditions for an effective cytotoxic effect during light induction [35]. Here, using laser scanning confocal microscopy, we analyzed the intracellular spatial distribution of **pz I** and **pz III** in glioma GL261 cells after 4 h of incubation (Figure 1). The co-localization of **pz I** or **pz III** with specific organelle markers was examined by comparing the signal distribution profiles in the corresponding fluorescent channels of organelle-specific dyes. The data revealed that the distribution profiles of **pz I** and **pz III** in glioma GL261 cells resemble those in fibrosarcoma MCA205 cells [27], predominantly in the Golgi apparatus and partially in the ER (Figure 1). Neither **pz I** nor **pz III** was detected in cellular compartments such as mitochondria, lysosomes, or nucleus. The localization of **pz I** and **pz III** in the ER indicates the ability of these photosensitizers to induce ICD when used in PDT [7,27,32,36,37].

#### 3.1.2. pz I-PDT and pz III-PDT Provide Efficient Photodynamic Activity against Glioma GL261 Cells

Next, we analyzed the photodynamic activity of **pz I** and **pz III** against glioma GL261 cells. A prior comprehensive assessment of the photophysical and photobiological properties of **pz I** and **pz III** in malignant and healthy brain cells [35,38] enabled us to determine the most effective concentrations of **pz I** and **pz III** for the induction of approximately 85–90% cell death in glioma GL261 cells. Employing such concentrations is essential for adhering to the gold standard evaluation of ICD in tumor models in vivo [39].

Indeed, the MTT cell viability assessment revealed that the application of **pz I** and **pz III** at concentrations of 2.8 µM and 1.7 µM, respectively, resulted in effective photodynamic activity against glioma GL261 cells under light irradiation of 20 J/cm^2^ (Figure 2A,B). We observed a gradual decrease in the percentage of viable glioma GL261 cells after **pz I**-PDT and **pz III**-PDT. Twenty-four hours after PDT, the fraction of dead cells reached 79 ± 1.1% (**pz I**-PDT) and 85 ± 2.3% (**pz III**-PDT).

Thus, the cell viability assessment showed that PDT based on **pz I** and **pz III** effectively killed glioma GL261 cells 24 h after light irradiation, pointing to the induction of ICD. To confirm our assumption, we then analyzed the dynamics of emission of the main ICD hallmarks after **pz**-based PDT, namely, DAMPs such as ATP and HMGB1.

#### 3.1.3. Analysis of DAMP Release from Glioma GL261 Cells after pz I-PDT and pz III-PDT Treatment

DAMPs act as adjuvants for antigen-presenting cells (e.g., DCs), facilitating their recruitment and maturation, the subsequent cross-presentation of antigenic peptides to CD8^+^ T cells, and the activation of a strong anti-tumor immune response [3,40,41]. Taking into account the high immunogenic potential of **pz I**-PDT and **pz III**-PDT [27], we analyzed the levels of several DAMPs, such as ATP and HMGB1, released from glioma GL621 cells at different time points after **pz**-based PDT.

The quantitation of ATP in the cell-free supernatants of **pz I**-PDT-induced glioma GL261 cultures revealed a significant increase in the ATP level by 14 h after light irradiation, which, at the time point of 24 h, exceeded the values in the untreated GL261 cells (“Before PDT”) by three-fold (Figure 3A). In contrast, the ATP release from glioma GL261 cells induced by **pz III**-PDT was less pronounced, and the ATP release tended to be the most intense immediately after PDT (0 h) and at earlier time points (1.5 h and 3 h) (Figure 3B).

The HMGB1 levels in the culture supernatants of the PDT-treated glioma GL261 cells were measured with a sensitive ELISA (Figure 4). The HMGB1 levels in the culture supernatants increased gradually after **pz I**-PDT (Figure 4A) and reached the maximum values of 119 ± 27 ng/mL at 24 h when it exceeded the values detected immediately after PDT by 2.7-fold (44.5 ± 7.4 ng/mL). Interestingly, the HMGB1 levels registered immediately after **pz III**-PDT (“0 h”) (7.2 ± 0.4 ng/mL) were significantly lower than those in the **pz I**-PDT group. A sharp increase in the HMGB1 levels in the culture supernatants after **pz III**-PDT was found 3 h after light irradiation, followed by a tendency to decrease by 24 h.

Thus, PDT based on **pz I** or **pz III** generates a special pattern of DAMP release from dying glioma GL261 cells. After **pz I**-PDT, glioma cell death was accompanied by a gradual increase in ATP and HMGB1 release during the 24 h after light irradiation. On the other hand, after **pz III**-PDT, the sharp increase in the HMGB1 levels was accompanied by an increase in ATP that was detected in the first hours after irradiation, although it was not statistically significant. These data suggest that ICD induced in glioma GL261 cells via **pz I**-PDT or **pz III**-PDT triggers different kinetics of DAMP emission, which might affect their immunogenic potential. Next, we analyzed the ability of glioma GL261 cells, stimulated with **pz I**-PDT or **pz III**-PDT, to activate an adaptive immune response and effectively prevent tumor growth in immunocompetent C57BL/6 mice.

### 3.2. Glioma GL261 Cells Induced by pz I-PDT or pz III-PDT Are Immunogenic in the Subcutaneous Tumor Prophylactic Vaccination Model

To examine the immunogenic potential of dead/dying glioma GL261 cells treated with **pz I**-PDT or **pz III**-PDT, we followed a gold standard protocol of subcutaneous tumor prophylactic vaccination [7,32] with some modifications (Figure 5).

Dead/dying glioma GL261 cells from **pz I**-PDT and **pz III**-PDT were used as prophylactic vaccines. First, we performed an additional flow cytometry analysis of cell death with Annexin V and Sytox Blue staining to determine the different cell death stages 24 h after PDT (Figure 5A). Annexin V fluorescence dye in combination with Sytox Blue enables the detection of cells exposing phosphatidylserine on the surface, which is typical of the early cell death stage and is also characteristic of apoptosis and some other regulated cell death modalities [42]. Here, we showed that in both **pz I**-PDT and **pz III**-PDT groups, the largest population is composed of Annexin V^+^/Sytox Blue^+^ cells, which correspond to a late stage of cell death that might have the strongest immunogenic potential (Figure 5A). Only 7.2 ± 1.3% (**pz I**-PDT) and 5.2 ± 3.3% (**pz III**-PDT) of the cells belonged to the Annexin V^+^/Sytox Blue^-^ population, which indicates an early stage of regulated cell death with phosphatidylserine exposed on the outer membrane without the disruption of plasma membrane integrity. Finally, 3.9 ± 0.8% and 3.9 ± 1.0% of cells in the **pz I**-PDT and **pz III**-PDT groups, respectively, were Annexin V^−^/Sytox Blue^+^, which demonstrates damage or destruction of the cytoplasmic membrane and death through non-regulated pathways [43]. This can also occur during the staining process.

Next, the dead/dying glioma GL261 cells after **pz I**-PDT or **pz III**-PDT were collected and injected subcutaneously twice in the left flank of immunocompetent C57BL/6 mice with an interval of 7 days. As negative controls (non-ICD), the mice were injected with PBS or with glioma cells subjected to accidental necrosis via several freeze–thaw cycles (F/T), which are characterized by weak immunogenic properties [26,28,44]. On day 8, after the last immunization, the mice were challenged in the opposite flank with viable glioma GL261 cells, and tumor growth was monitored for 28–32 days (Figure 5B). The glioma GL261 cells treated with **pz I**-PDT or **pz III**-PDT acted as effective tumor prophylactic vaccines capable of activating an adaptive immune response in mice (Figure 5C,D). By the end of the observation period, the survival rates of mice in the **pz I** and **pz III** groups were 90% and 70%, respectively (Figure 5C). The tumor volumes were 65.32 ± 51.28 mm^3^ (**pz I**) and 139.9 ± 53.7 mm^3^ (**pz III**), which were significantly smaller than in the control PBS group (1360 ± 26 mm^3^) (Figure 5D). The survival of mice immunized with F/T glioma GL261 cells was about 60% by day 15 after the injection of viable glioma cells, and none of them were alive by day 28. This result is consistent with previous findings, providing strong evidence of the poor immunogenic properties of cancer cells dying via accidental necrosis [26,28,44]. All of the mice in the PBS group died 6–8 days after inoculation with glioma GL261 cells.

Overall, these data confirm that PDT based on **pz I** or **pz III** effectively induces ICD in glioma GL261 cells that triggers an adaptive immune response and protection from tumor growth in vivo in a subcutaneous tumor prophylactic vaccination model.

### 3.3. DC Vaccines Loaded with Lysates of Glioma GL261 Cells after pz I-PDT or pz III-PDT Protect Mice in an Orthotopic Intracranial Glioma Model

As we demonstrated that the use of whole dead/dying glioma GL261 cells as prophylactic vaccines provide effective protection against challenges with viable glioma cells in mice, the final part of our study aimed to validate the immunogenic potential of DC vaccines loaded with the lysates of glioma GL261 cells that underwent ICD after **pz I**-PDT or **pz III**-PDT treatment.

Immunocompetent syngeneic C57BL/6 mice received two intraperitoneal vaccinations with DCs loaded ex vivo with the lysates of glioma GL261 cells treated with **pz I**-PDT or **pz III**-PDT (Figure 6A). As negative controls, the mice were vaccinated with DCs pulsed with the lysates of glioma GL261 cells subjected to several F/T cycles or with PBS (Figure 6A). One week after the last immunization, all of the mice were inoculated intracranially with viable GL261 glioma cells (2 × 10^4^ cells per 20 g of mouse weight). The glioma cells were inoculated 2 mm lateral and 2 mm posterior to the bregma and 3 mm below the dura mater. These coordinates corresponded to one of the nuclei of the hypothalamus [45]. This means that tumor progression is well correlated with disturbances in the thalamic nuclei and surrounding structures, and these disturbances are directly reflected in neurological deficits because they are associated with the most important cognitive functions in mice.

The development of neurological deficit symptoms, the survival rate of mice, and the MRI monitoring of tumor progression were followed for more than 50 days after the intracranial inoculation with viable glioma GL261 cells (Figure 6B–D).

We found that prophylactic vaccination with DCs loaded with accidentally necrotic glioma GL261 cells (i.e., the F/T group, non-ICD) does not protect the mice against challenges with viable GL261 cells, with a 60% mortality by day 26 and a 100% mortality on day 32 (Figure 6B). The non-immunogenic nature of DC vaccines pulsed with accidentally necrotic glioma GL261 cells was accompanied by the development of a severe neurological deficit in the mice (Figure 6C) and significant changes in the brain tissue morphology according to the MRI images (Figure 6D). On the other hand, the use of DCs pulsed with the lysates of glioma GL261 cells after **pz I**-PDT or **pz III**-PDT significantly protected the mice from tumor growth and prolonged their survival compared to the negative controls (i.e., PBS or DC-F/T groups). By the end of the observation period (day 52), the survival rates of the mice in the DC-**pz I**-PDT and DC-**pz III**-PDT groups were 45.5% and 73%, respectively. Interestingly, the survival of the mice vaccinated with DCs loaded with the lysates of **pz III**-PDT treated glioma GL261 cells was not statistically different from the survival of the mice injected with PBS (day 52), but the neurological scores during the period of observation (52 days) were remarkably different (Figure 6C). On day 20 after viable glioma cell inoculation, the neurological deficit in the mice that were immunized with DCs loaded with the lysates of **pz-I**-PDT-treated glioma GL261 cells was significantly lower than in the control PBS group. The use of DC vaccines pulsed with **pz-III**-PDT glioma-treated cells increased the survival rate from day 26 after the inoculation of viable glioma cells and, according to the analysis of the MRI images on day 23 (Figure 6D), decreased tumor growth in comparison to the DC-F/T group. An analysis of the MRI tomograms revealed that the average tumor volumes in the control PBS and the DC-F/T groups were 0.51 ± 0.06 cm^3^ and 1.258 ± 0.328 cm^3^, respectively. The tumors were mainly located to the left of the bregma, along the sagittal suture, and spread superficially over the cerebral cortex. In the mice that were vaccinated with DCs loaded with glioma GL261 lysates after **pz I**-PDT treatment, the tumor site had no clear boundaries and averaged 0.11 ± 0.09 cm^3^; an active deep penetration of tumor cells into nervous tissue was also observed. The tumor volume in the mice in the DC-**pz III**-PDT group averaged 0.375 ± 0.375 cm^3^. The tumors were mainly in the second quarter of the right cerebral hemisphere and had clear boundaries.

Overall, our data indicate that prophylactic vaccination with DCs loaded with glioma GL261 lysates treated with **pz I**-PDT or **pz III**-PDT is an effective inducer of protective anti-tumor immunity against glioma growth in the orthotopic intracranial mouse model.

## 4. Discussion

Cancer remains one of the most serious problems for public health. The high incidence rate of cancer (World Health Organization https://www.who.int/news-room/fact-sheets/detail/cancer, accessed on 15 December 2022), risks of secondary tumors, low effectiveness of conventional therapy, and disappointing prognosis for patients justify the need for the development of novel breakthrough approaches for the treatment and timely diagnosis of cancer. This is particularly important for aggressive tumor types, such as brain tumors (gliomas) [46,47,48].

To date, PDT is considered a prospective adjuvant therapy for cancer, including brain tumors [16,18,19,26]. At the same time, a number of studies have shown that the manifestation of immunogenicity by dying tumor cells is the most important factor that increases the effectiveness of anticancer therapy. The induction of ICD leads to the emission of specific pattern of DAMPs, which act as danger signals and as adjuvants to activate the immune system [3,7]. However, due to specifics of the tumor’s profile, the photodynamic activity and photobiological properties of the photosensitizers used, and the variety of photodynamic treatment modes, tumor cell death can follow immunogenic or non-immunogenic pathways, which determines the rationality of PDT and the degree of its effectiveness for the treatment of cancer [16,18].

Our study was focused on photosensitizers from the tetra(aryl)tetracyanoporphyrazines group (**pz**). The advantage of **pz** compared to clinically approved “classic” photosensitizers is their strong cytotoxic effects against several types of tumor cells, combined with the possibility of using them as viscosity sensors for the real-time assessment of tumor cell response after light irradiation. They could be considered as potential photodynamic agents for dosimetry-assisted PDT (i.e., viscosity sensors) to assess local intracellular viscosity and to visualize viscosity changes driven by PDT treatment [22,23,24,25,49,50]. Moreover, according to our previous in vitro and in vivo studies, these original photodynamic dyes, including **pz** with 9-phenanthrenyl (**pz I**) and 4-(4-fluorobenzyoxy)phenyl (**pz III**) groups in the aryl frame of the macrocycle, have great immunogenic potential against murine fibrosarcoma cells [27]. This strengthens their positions as prospective photosensitizers for PDT for cancer. In this study, we assessed the immunogenic properties of PDT based on **pz I** or **pz III** against glioma GL261 cells in vitro and analyzed whether glioma GL261 cells stimulated with **pz**-based PDT can activate the adaptive immune response and protect from tumor (i.e., glioma) growth in a tumor prophylactic vaccination model. Moreover, we examined the effectiveness of DC vaccines pulsed with the lysates of glioma GL261 cells subjected to **pz I**-PDT or **pz III**-PDT as effective inducers of anti-tumor immunity in an intracranial orthotopic glioma model in immunocompetent mice.

We have recently shown the considerable accumulation of **pz I** and **pz III** in glioma cells [35]. We detected an active penetration of **pz** in glioma cells after 2 h of incubation and strong cytotoxic effects during light irradiation [35]. At the same time, the ability of PDT to induce cancer cell death through the immunogenic pathway depends on the spatial distribution of the photosensitizer in the cells [16,30,31,33,34]. Herein, we have shown that after **pz I** and **pz III** penetrate into glioma GL261 cells, they localize predominantly in the Golgi apparatus and partially in the ER. Our findings are consistent with previously published studies indicating that the localization of photosensitizers in the ER can lead to ER stress after PDT, which is closely associated with the efficient induction of ICD [3,27,32,33]. Our results also demonstrate that PDT based on **pz I** and **pz III** contributes to the emission of ICD hallmark molecules from dying glioma GL261 cells. In the case of **pz I**-PDT, glioma cell death was accompanied by a gradual increase in the release of two crucial DAMPs, ATP and HMGB1, during the 24 h after light irradiation. On the other hand, **pz III**-PDT leads to an increase in the HMGB1 and ATP levels, mainly during the first 3 h after irradiation. It is conceivable that ATP released in the first hours after **pz III**-PDT treatment can be hydrolyzed to adenosine [51]. The observed differences in the DAMP profiles of the two photosensitizers are presumably related to different mechanisms of glioma GL261 cell death triggered by **pz I**-PDT and **pz III**-PDT. Our interest is fueled by previous data indicating that **pz I** and **pz III** can induce different cell death modalities in cancer cells during PDT, including a mixed cell death phenotype [27]. In fact, this is an attractive strategy that might have great potential to overcome the frequently observed resistance of cancer cells to a single cell death type [18,26,40,52,53,54]. Our in vitro findings were confirmed in vivo, demonstrating that glioma GL261 cells stimulated with **pz I**-PDT or **pz III**-PDT served as potent vaccines in a tumor prophylactic vaccination model, which points to the activation of an adaptive anticancer immune response. The C57BL/6 mice immunized with dying/dead GL261 glioma cells were well protected against tumor growth at the challenge site and demonstrated high overall survival rates (90% for **pz I** and 70% for **pz III**), much higher than the control mice.

Vaccination with DCs pulsed with whole glioma lysates has been drawing a lot of attention as a prospective immunotherapeutic approach for gliomas [55,56]. One of its main advantages is that it reduces the likelihood of immune escape variants because it does not depend on specific tumor cell phenotypes, but allows for the immune system to target a wide range of tumor antigens, including any neoantigens that may be presented [57,58,59,60,61]. Following this idea, the final step of our study assessed the utility of using DC vaccines pulsed with the lysates of glioma GL261 cells after ICD induction by **pz I**-PDT or **pz III**-PDT as effective inducers of adaptive anti-tumor immunity in an intracranial orthotopic glioma model in immunocompetent C57Bl/6 mice. Our findings show that the lysates obtained from glioma GL261 cells treated with **pz I**-PDT or **pz III**-PDT have promising prospects for the generation of personalized DC vaccines to provide better protection against glioma growth and the development of neurological deficit, and to prolong survival.

The use of **pz I**-PDT leads to a more uniform release of DAMPs and their greater accumulation in the cell supernatants 24 h after PDT. On the other hand, **pz III**-PDT demonstrates stronger protective properties after the immunization of mice, suppressing tumors, prolonging survival, and activating a more pronounced immune response in both syngeneic and orthotopic glioma mouse models. These results provide strong rationale for further research on both **pz** photosensitizers to pave the way for enhancing the therapeutic potential of photodynamic therapy and DC vaccines for experimental glioma treatment.

## 5. Conclusions

This study characterized, in vitro and in vivo, the immunogenic potential of two photoactive dyes from the tetracyanotetra(aryl)porphyrazine group, **pz I** and **pz III**, against experimental glioma. Both photosensitizers can act as efficient ICD inducers in PDT-treated glioma GL261 cells in vitro and provide strong protection against tumor growth in a prophylactic subcutaneous glioma vaccination model. More importantly, DCs pulsed with the lysates of dying/dead glioma GL261 cells after **pz I**-PDT or **pz III**-PDT served as effective prophylactic vaccines that activated anti-tumor immunity, significantly reduced the rate of tumor growth, prolonged mouse survival, and protected against the development of neurological deficit. Our findings potentially broaden the prospects for the development of effective immunotherapeutic strategies based on PDT and DC vaccines loaded with the whole tumor lysates in the treatment of brain tumors.

## Figures and Tables

**Figure 1 pharmaceutics-15-02430-f001:**
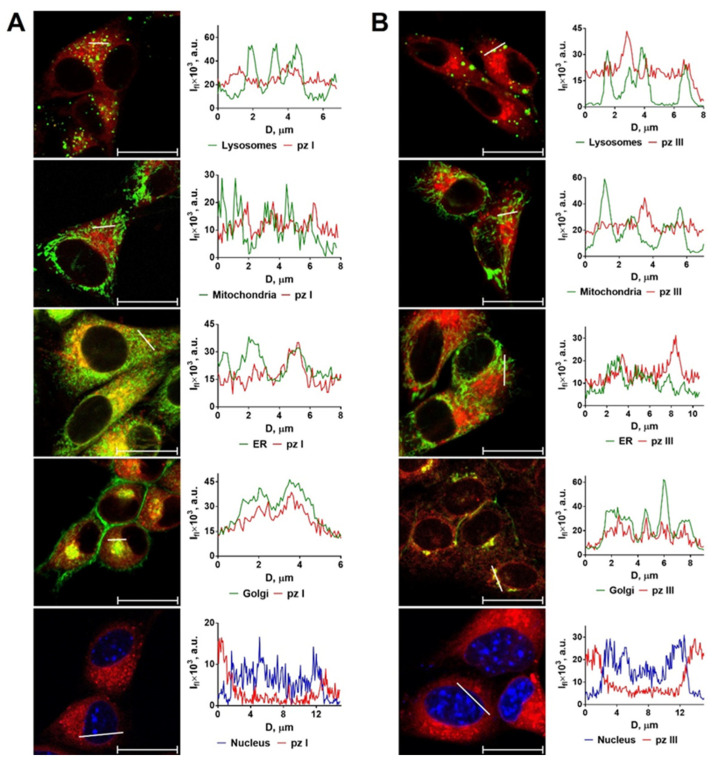
Confocal images of the intracellular spatial distribution of pz I (**A**) and pz III (**B**) in glioma GL261 cells. **Pz I** and **pz III** are localized predominantly in the Golgi apparatus and partially in the ER after 4 h of incubation. Importantly, **pz I** and **pz III** were not detected in mitochondria, lysosomes, or nucleus. The following dyes were used for co-localization studies: LysoTracker Green (DND-26) for lysosomes, MitoTracker Green for mitochondria, ERTracker for ER, and DAPI for nuclei. In the graphs: red line represents the fluorescence intensity of **pz**; green line represents the fluorescence intensity of dye in the indicated cellular compartment (I_fl_, a.u. ×10^3^); D is the distance along the specified segment (μM). Scale bars: 20 μM.

**Figure 2 pharmaceutics-15-02430-f002:**
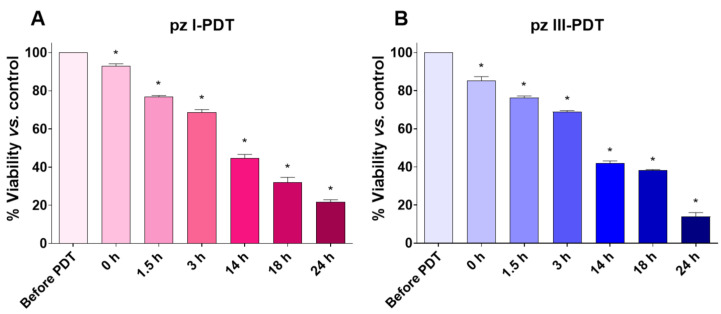
Photodynamic activity of pz I and pz III against glioma GL261 cells. The GL261 cells were incubated with **pz I** (2.8 µM) or **pz III** (1.7 µM) in a serum-free medium for 4 h and then exposed to light irradiation at 20 J/cm^2^ using an LED light source (λex 630 nm; 20 mW/cm^2^). (**A**,**B**) Cell viability analysis using MTT assay at different time points after PDT. Dashed line: percent viability was 100% in all cases. “Before PDT”: untreated glioma cells. “0 h”: cell viability measurements immediately after PDT. *: versus viable glioma GL261 cells, *p* < 0.05, Mann–Whitney test. All data represent the mean values ± standard error of the mean (SEM) of three independent experiments.

**Figure 3 pharmaceutics-15-02430-f003:**
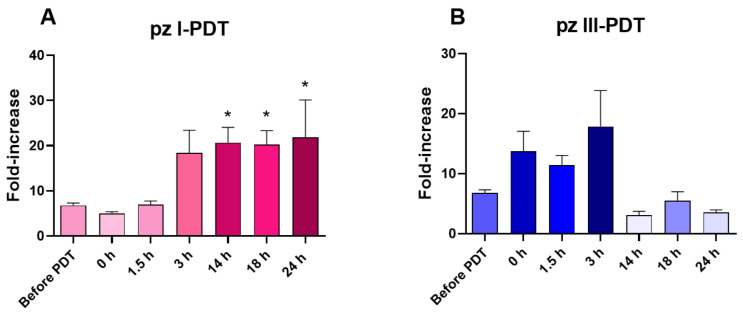
Analysis of ATP release from glioma GL261 cells after pz I-PDT (**A**) **and pz III-PDT** (**B**). ATP release in the cell-free supernatants was measured and plotted as fold increase relative to medium with 2% FBS. “Before PDT”: ATP measurements from untreated glioma cells. ”0 h”: ATP measurements from glioma cells immediately after PDT. *: versus “Before PDT” group, *p* < 0.05, Mann–Whitney test. The data represent the mean values ± standard error of the mean (SEM) of three independent experiments.

**Figure 4 pharmaceutics-15-02430-f004:**
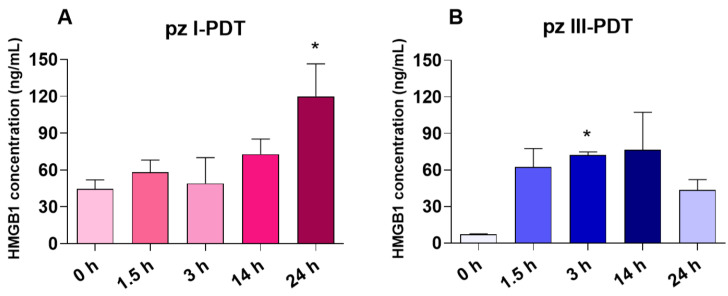
Analysis of HMGB1 release from glioma GL261 cells after pz I-PDT (**A**) **and pz III-PDT** (**B**). Before PDT: HMGB1 measurements from untreated glioma cells. “0 h”: HMGB1 measurements from glioma cells immediately after PDT. HMGB1 was measured in the cell-free supernatants and represents the mean values of three independent experiments. *: versus “0 h” group, *p* < 0.05, Mann–Whitney test. The data represent the mean values ± standard error of the mean (SEM) of three independent experiments.

**Figure 5 pharmaceutics-15-02430-f005:**
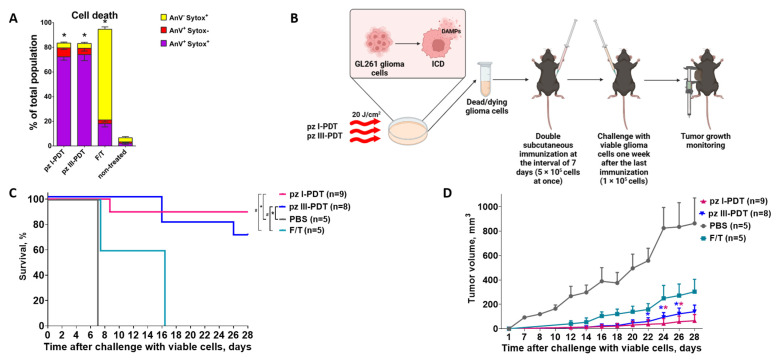
Immunogenicity assessment of dead/dying glioma GL261 cells from pz I-PDT or pz III-PDT in the subcutaneous tumor prophylactic vaccination model. (**A**) Cell death analysis based on Annexin V-FITC and Sytox Blue staining 24 h after **pz I**-PDT or **pz III**-PDT; “F/T”—positive control of cell death and cells subjected to freeze–thaw cycles; “non-treated”—viable glioma GL261 cells. All data represent the mean values ± standard error of the mean (SEM) of three independent experiments. The statistical difference (*): versus non-treated cells, *p* < 0.05, multiple *t*-test. (**B**) Subcutaneous tumor prophylactic vaccination of immunocompetent C57BL/6 mice with photoinduced glioma GL261 cells. The cells were exposed to **pz I**-PDT or **pz III**-PDT and incubated overnight for 24 h as described in Materials and Methods (Section 2.2). The mice were subcutaneously inoculated twice in the left flank with dead/dying glioma GL261 cells (5 × 10^5^ cells) with an interval of 7 days. For the negative control, the mice were injected with the same quantity of glioma GL261 cells subjected to freeze–thaw (F/T) cycles (i.e., non-ICD) or with PBS only. On day 8, after the last immunization, the mice were challenged subcutaneously on the opposite flank with viable glioma GL261 cells (5 × 10^5^ cells). Tumor growth at the challenge site was measured over the following 28–32 days. (**C**) Tumor appearance at the challenge site is represented as a Kaplan–Meier curve. The statistical difference (*): versus PBS; #: versus F/T, *p* < 0.01, long-rank Mantel–Cox test. (**D**) Dynamics of tumor growth at the challenge site after inoculation of viable glioma GL261 cells; *: versus PBS group, *p* < 0.05, multiple *t*-test.

**Figure 6 pharmaceutics-15-02430-f006:**
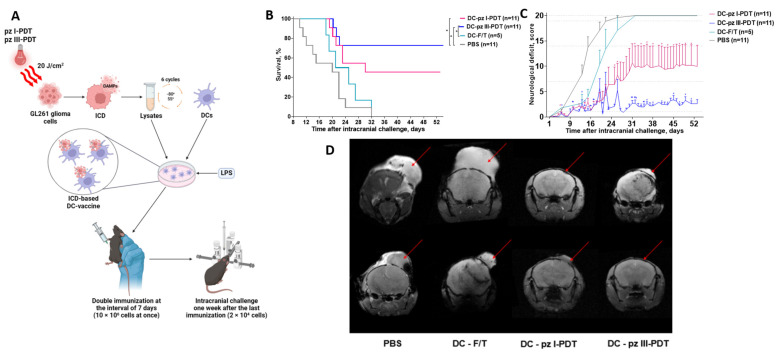
Efficiency of prophylactic DC vaccines pulsed with lysates of glioma GL261 cells killed by pz I or pz III PDT in an orthotopic intracranial glioma mouse model. (**A**) A scheme of glioma prophylactic DC vaccination. The immunocompetent C57BL/6 mice were injected intraperitoneally twice 7 days apart with a suspension of 1 × 10^6^ DCs loaded with photoinduced glioma GL261 cell lysates and hyper-stimulated via *E. coli* lipopolysaccharide (LPS). One week after the last immunization, viable glioma GL261 cells were inoculated intracranially at 2 × 10^4^ cells per 20 g of mouse weight. (**B**) Survival of mice vaccinated with the DCs and challenged with viable glioma GL261 cells. *: versus PBS; #: versus freeze–thawed (F/T) cells, Mantel–Cox test, *p* < 0.05, n = 5–11 per group. (**C**) Dynamics of neurological deficit development in mice after intracranial inoculation of viable glioma GL261 cells. *: versus PBS; #: versus F/T, Wilcoxon test, *p* < 0.05, n = 5–11 per group. (**D**) Morphological changes in mouse brain tissue after intracranial inoculation of glioma GL261 cells. Representative MRI images of mouse brains on day 23 post-challenge with viable glioma GL261 cells. The multi gradient echo multi slices were obtained with a high-field magnetic resonance tomograph, Agilent Technologies DD2-400 9.4 T (400 MHz) (Cheadle, UK), and the following parameters were assessed: TR = 1000 ms; TE = 1.49 ms; 6 echoes; FOV, 20 mm × 20 mm; matrix, 256 × 256; slice thickness, 1 mm; 15 slices; 17 min; and 4 sec scanning time.

## Data Availability

Data will be provided upon request.

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
