# Peer review of "Dendritic Cells Pulsed with Tumor Lysates Induced by Tetracyanotetra(aryl)porphyrazines-Based Photodynamic Therapy Effectively Trigger Anti-Tumor Immunity in an Orthotopic Mouse Glioma Model"

_pharmaceutics, 2023, doi:10.3390/pharmaceutics15102430_

Round 1

Reviewer 1 Report

The manuscript entitled “Dendritic cells pulsed with tumor lysates induced by tetracyanotetra (aryl) porphyrazines-based photodynamic therapy effectively trigger anti-tumor immunity in an orthotopic mouse glioma model” by Tikhon S Redkin et al. provides valuable insights into the role of immunogenic cell death (ICD) in anticancer therapy and the potential of photoactive dyes in inducing ICD. The study emphasizes the role of ICD in activating specific antitumor immune responses, which leads to the elimination of tumor cells and the formation of long-term immunological memory. The research focuses on characterizing the abilities of two photoactive dyes, from the tetracyanotetra (aryl) porphyrazine group, to induce ICD in experimental glioma both in vitro and in vivo. The findings suggest that these photosensitizers can act as efficient ICD inducers and provide strong protection against tumor growth. Furthermore, dendritic cell (DC) vaccines pulsed with lysates of glioma GL261 cells pre-treated with these photosensitizers can act as effective inducers of adaptive anti-tumor immunity in the intracranial orthotopic glioma mouse model. This potentially broadens the prospects for the development of effective immunotherapeutic strategies based on PDT, ICD, and DC vaccines for the treatment of brain tumors. The research paves the way for the development of innovative immunotherapeutic strategies for treating brain tumors.

Minor Comments:

1.       The representation of Figures 5 and 6 lacks uniformity. It would be beneficial for the readers if the figures followed a consistent format throughout the manuscript.

2. Panels A and C of Figure 5 are missing statistical data. It is crucial to include this information to provide a comprehensive understanding of the results and to validate the findings.

Author Response

We thank the Reviewer#1 for the comments and appreciation of the manuscript.
Please see the attachment.

Reviewer 2 Report

The manuscript lacks novelty. The induction of ICD is a well-known process that leads to the immune rejection of tumors; therefore, the provided results are predictable.

In Section 3.1, the authors should provide additional evidence of DAMPs, in addition to measuring ATP and HMGB1.

Ideally, the authors should also assess the infiltration of T cells (e.g., using IHC or RNA seq) and other immune cells in the tumor tissue. They should thoroughly dissect and photograph all tumors and record their weights.

It is unclear how the authors prepared the F/T control with necrotic cells.

Regarding lines 456 to 460, the writing appears to be inconsistent with the results displayed in Figure 6c. Please clarify or double-check this section for accuracy.

ok

Author Response

We’d like to thank the Reviewer#2 for the comments. Please see the attachment.

Reviewer 3 Report

The research work “Dendritic cells pulsed with tumor lysates induced by tetracyanotetra(aryl)porphyrazines-based photodynamic therapy effectively trigger anti-tumor immunity in an orthotopic mouse glioma model” focuses on team has characterized the abilities of two photoactive dyes to induce ICD in experimental glioma in vitro and in vivo. They proved that that dendritic cell (DC) vaccines pulsed with lysates of glioma GL261 cells pre-treated with pz-I-PDT or pz-III-PDT can act as effective inducers of adaptive anti tumor immunity in the intracranial orthotopic glioma mouse model. The research work is novel and as per scope of journal. The minor revisions/justifications are as follows.

The comments are:

1.       Line 45-47: The abstract last few lines and conclusion line are same, please rewrite:

2.       Please cite methods, 2.1 Glioma cell line, 2.3 Subcellular distribution of pz I and pz III in glioma cells, 2.4 Cell death assay by MTT and flow cytometry, 2.5 Analysis of DAMPs release: HMGB1 and ATP, and 2.6 Mouse experiments (All subpart or wherever relevant), in cases where method used from literature and not developed inhouse.

3.       Why figure 1, figure 2 etc is highlighted in yellow in results?

Author Response

We’d like to thank the Reviewer for the comments. Please see the attachment.
